# MicroRNAs as a Potential Quality Measurement Tool of Platelet Concentrate Stored in Blood Banks—A Review

**DOI:** 10.3390/cells8101256

**Published:** 2019-10-15

**Authors:** Jersey Heitor da Silva Maués, Caroline de Fátima Aquino Moreira-Nunes, Rommel Mário Rodriguez Burbano

**Affiliations:** 1Laboratory of Human Cytogenetics, Institute of Biological Sciences, Federal University of Pará, Belém, PA 66075-110, Brazil; rommel@ufpa.br; 2Laboratory of Molecular Biology, Ophir Loyola Hospital, Belém, PA 66063-240, Brazil; 3Laboratory of Pharmacogenetics, Drug Research and Development Center (NPDM), Federal University of Ceará, Fortaleza, CE 60430-275, Brazil; 4Christus University Center-Unichristus, Faculty of Biomedicine, Fortaleza, CE 60192-345, Brazil

**Keywords:** platelet concentrate, biomarkers, injury, microRNA, blood banks

## Abstract

Background: Platelet concentrate (PC) is one of the main products used in a therapeutic transfusion. This blood component requires special storage at blood banks, however, even under good storage conditions, modifications or degradations may occur and are known as platelet storage lesions. Methods: This research was performed on scientific citation databases PubMed/Medline, ScienceDirect, and Web of Science, for publications containing platelet storage lesions. The results obtained mainly reveal the clinical applicability of miRNAs as biomarkers of storage injury and as useful tools for a problem affecting public and private health, the lack of PC bags in countries with few blood donors. The major studies listed in this review identified miRNAs associated with important platelet functions that are relevant in clinical practice as quality biomarkers of PC, such as miR-223, miR-126, miR-10a, miR-150, miR-16, miR-21, miR-326, miR-495, let-7b, let-7c, let-7e, miR-107, miR-10b, miR-145, miR-155, miR-17, miR-191, miR-197, miR-200b, miR-24, miR-331, miR-376. These miRNAs can be used in blood banks to identify platelet injury in PC bags. Conclusion: The studies described in this review relate the functions of miRNAs with molecular mechanisms that result in functional platelet differences, such as apoptosis. Thus, miRNA profiles can be used to measure the quality of storage PC for more than 5 days, identify bags with platelet injury, and distinguish those with functional platelets.

## 1. Introduction

Platelet concentrate (PC) is one of the major blood products used in a therapeutic transfusion. This blood component requires special storage, however, even under good storage conditions, modifications or degradation may occur and are known as storage lesions [1]. These PC modifications include morphological and physiological changes, platelet activation, changes in membrane glycoproteins and proteolysis, and expression of platelet surface receptors. These changes may alter the structure and function of platelets and are referred to as the platelet storage lesion (PSL) [2]. One reason for this loss of quality is due to mitochondrial dysfunction, and apoptosis and ultimately viability loss during storage [3].

A serious problem that affects public and private health is the low rates of PC bags that commonly affect countries with few blood donors, such as Brazil. In addition to the lack of donors, the main difficulty is that after five days of storage in the blood bank, all unused PC bags are discarded as a result of storage lesions [4], and on the sixth day PCs may be missing from the stock, causing serious problems to hospitals dependent on a blood bank. However, many of these PC bags may still contain functional platelets, since the kinetics of cellular aging is influenced by the age, type of feeding, and exposure to environmental agents to which blood donors may have undergone [5,6]. Biomarkers of PC quality are therefore highly sought after in blood bank governance.

A proposed mechanism based on the molecular regulation of microRNAs (miRNAs) has been shown to be relevant for understanding storage lesions, as platelets when stored, continue to translate mRNA proteins [7]. The biological role of miRNA is mainly linked to its ability to act together and mediate sequence-specific regulation by repressing or degrading mRNAs at specific non-translated binding sites [8,9]. This review addresses the major advances in the identification of platelet miRNAs that are important in the field of blood components transfusion and associating the role of miRNAs as a tool to measure the quality of platelets stored in blood banks.

## 2. Platelet Storage Lesions

The first changes in platelets during PC storage were mentioned in 1957 by Mustard et al., who described the influence of platelet number collection techniques during the blood storage procedure [10].

Murphy et al., in 1971, introduced the term Platelet Storage Lesion (PSL) related to the following observations: lactate accumulation, discoid-to-sphere transformation, and reduced Adenosine diphosphate (ADP) responsiveness for platelet aggregation [11]. Later, studies described that the mechanisms responsible for PSL are multifactorial and include the methods of collection, processing, storage, manipulation after collection, and shelf life [10,12].

PC is one of the major blood products used in therapeutic transfusion and is stored for up to five days at room temperature 22 ± 2 °C with gentle and continuous shaking. Platelets need this special storage because when stored at room temperature for transfusion for more than five days, they undergo changes collectively known as PSL [13]. Longer duration of platelet storage has not been recommended due to the possible risk of bacterial contamination [2,14].

Most of these PC storage lesion changes are related to platelet activation and include exposure to foreign surfaces, trauma, low pH, agonists (such as thrombin and ADP), shear stress, anaerobic glycolysis that results in lactate accumulation, among others [12,15,16]. In addition, the occurrence of platelet apoptosis is characterized by the most common events such as cytoskeletal protein degradation, ATP depletion, caspase activation, and decreased mitochondrial membrane potential [12,17,18]. More recent studies revealed potential miRNAs as efficient biomarkers to identify the molecular mechanisms by which storage conditions for more than 5 days result in functional differences in human platelets stored in blood banks [7,19,20].

## 3. Platelet MicroRNAs

Early studies on the existence of platelet miRNAs have raised the hypothesis of their involvement in platelet activation that was observed from the analysis of pre-mRNA splicing and their processing during translation, where activated platelets showed changes in their transcriptome and proteome in response to activation, which led to believe that these events occurred in a nucleus-free environment, that is, in the cytoplasmic environment of activated human platelets [21,22].

In 2006, Garzon et al. published a study that became known as platelet function miRNA fingerprints, where it was reported that platelets, even without nuclei and genomic DNA, had several types of coding and non-coding RNAs. In this study, miRNAs with a downregulated profile such as (miR-10a, miR-126, miR-106, miR-10b, miR-17, and miR-20) involved in megakaryocytopoiesis were identified [23] (Table 1).

The first miRNA expression profile in human platelets was reported as part of a test study, where differentially expressed miRNAs were identified in patients with polycythemia vera. In this study, it was observed that miR-26b expression was significantly higher in the platelets of patients with this disorder compared to healthy controls [41].

Performing a membrane array-based miRNA analysis, Kannan (2009) confirmed the existence of 52 apoptosis-associated platelet miRNAs [24], most of these miRNAs are listed in Table 1. The expression of a large number of platelet miRNAs (total of 219) was first reported by Landry et al. in 2009 using microarray profiles [25]. However, the microarray limitation allowed the identification of only known miRNA sequences in the miRBase database (http://www.mirbase.org).

The comparative study by Nagalla et al. (2011), correlated with Landry et al. (2009) earlier findings, using a strategy with a broad miRNA genomic profile in 19 healthy donors, where a subset of 284 miRNAs in platelets was identified, confirming miR-223 as the most abundant. Another study on the characterization of human platelet miRNAs identified a total of 281 transcripts, of which six (miR-15, miR-339, miR-365, miR-495, miR-98, and miR-361) (Table 1) were more differentially expressed [26]. It has also been shown how miRNA-mRNA co-expression profiles correlated with platelet reactivity, consolidating the importance of the miRNA regulatory role in modulating platelet mRNA translation [27].

Next-Generation Sequencing (NGS) technique made it possible to discover new sequences and increasing the total number of platelet-expressed miRNAs to more than double the initial findings, totaling 490 miRNAs. In human platelets, one of the most abundant miRNA families has been described as members of the let-7 family, which account for 48% of platelet miRNA content [28]. The existence of let-7 family members (let-7a, -7c, -7e, -7f, -7g, and -7i) (Table 1) has been previously confirmed on platelets during storage [24]. Subsequent studies also showed different results identifying the miR-127/miR-320a [7] and miR-548 [29] family that were shown to be abundant during platelet storage. These findings show that different sequencing technologies, combined with bioinformatics tools and pipelines, yield different results and show how the platelet miRNA repertoire is relatively diverse and complex.

One study showed miRNAs involved in regulating platelet apoptosis gene expression, where their mitochondrial DNA plays a key role in regulating apoptosis [30]. A second study demonstrated a significant increase in miR-326 expression in apheresis-derived platelets that were stored in vitro. The *BCL*-2 family regulator composed of the anti-apoptotic *Bcl-xL* has been identified as a miR-326 target [31].

The first complete miRNA sequencing (miRNome) of blood bank stored platelet concentrate was obtained with NGS and revealed a gradual decrease of the most abundant miRNAs in PC. In addition, an inverse relationship of expression between miR-127 and miR-320a was observed depending on platelet storage time. This relationship enables the identification of PC bags that may still have physiologically normal (not activated) platelets [7].

In order to shape mRNA profile and murine platelet function, Rowley et al. (2016) used a molecular alteration system to delete Dicer1, as described in their study [32], which resulted in increased expression of αIIbβ3 integrins on the surface of these cells, leading to increased platelet reactivity [32,42]. The authors suspected that miRNAs would be regulating these integrins and tested their hypotheses using 3′ UTR (untranslated region) luciferase reporter assays confirming modulation of these mRNAs by miRNAs (miR-326, miR-128, miR-331, and miR-500), in megakaryocyte and potentially in platelets in the bloodstream or during storage [32].

In many countries with few blood donors, it is common during PC storage, even under optimal storage conditions, platelets begin to lose their quality, which is due to mitochondrial dysfunction [43]. Using bioinformatics analyzes one study showed *ATP5L* encoding mRNA as a potential target for miR-570-3p, which was identified with high levels during platelet storage, being able to clarify the mechanisms of mitochondrial dysfunction of this cell by miRNA regulation by mRNAs. The miR-570 (Table 1) have been proposed as a reliable biomarker for the quality and viability of stored platelets [29].

Release of miRNA-rich microparticles (MPs) is mainly triggered by factors such as shear stress and activation, which may also modify miRNA expression profiles during storage in blood banks [44,45]. A study by Gibbings et al. (2009) describes a miRNA uptake pathway by MPs, which reinforces our hypothesis about the release of miRNA-rich microparticles during activation. The main pathway called exosomal occurs through the lumen sprouting of multivesicular bodies that are released by exocytosis [46]. MiRNAs in circulating MPs were transferred in vivo to tumor cells to exert regulatory functions over gene expression to modulate tumor progression [47]. Several studies currently support the release and transfer of encapsulated miRNAs in extracellular MPs formed by the cell membrane detachment process, including human platelets [48,49,50].

A more recent study published integrated NGS data applying an OMIC approach, that is a technology used to explore the roles and functions of molecules such as miRNAs, with bioinformatics tools and analysis, which identified the largest amount of differentially expressed miRNAs caused by prolonged storage for more than five days in blood banks [19]. Therefore, extensive research has been conducted to identify the association of miRNA as a quality measurement tool for blood bank stored platelets. Major publications on platelet miRNAs and their relationship to platelet storage lesions are on a timeline designed to illustrate these events, (Figure 1).

## 4. MicroRNAs as a Blood Bank Platelet Quality Measurement Tool

### 4.1. Downregulated miRNAs on Stored PC Bags

The miRNA differential expression profile is a useful tool for identifying changes in platelet physiology during storage [3]. Studies investigating platelet storage conditions indicate that miRNA profiles identified at different time intervals are quite different [19,24,33]. These studies revealed a link between microRNA profiles and subsequent platelet reactivity, thus suggesting the important role played by post-transcriptional regulation during storage.

Since platelets are small anucleate cells derived from the cytoplasmic fragmentation of megakaryocytes [51], they are unable to synthesize new miRNAs and therefore it is expected that the majority of miRNA expression levels of platelets decreases over time when platelets are stored at room temperature [7,24,30]. However, specific levels of miRNA expression may increase under the same conditions [19]. All RNAs have distinct inherent half-lives that dictate their level of accumulation and miRNAs would be expected to follow a similar principle [52]. However, the half-life of a miRNA molecule varies [53] and may change rapidly or remain stable for more than 12 days [54].

Platelets are able to synthesize a limited amount of proteins from resident mRNAs [55], along with other processes such as modulating miRNA expression [56], in platelets extensive pre-mRNA maturation occurs as responses to stimuli activators, representing a mechanism for post-transcriptional control of the proteome arrangement in these anucleated cell fragments. Still, studies have shown that platelet proteomic data correlated well with transcriptome, as about 69% of secreted proteins were detected at the mRNA level, meaning that not all proteins were synthesized again [57,58].

Platelet physiological changes are related to both, decrease and the increase in the number of miRNAs during storage [19,20]. The miRNAs differential expression profile is able to predict and monitor how the half-life of this molecule varies and undergoes rapid stability changes during storage days. The miRNAs listed in Table 1 are related to the most studied functions related to platelet storage lesions.

These miRNAs have been identified with inactivation and other associated functions such as reactivity, hemostasis, differentiation and apoptosis. For example, platelet activation-related miRNAs have been identified with differential expression analysis including miR-10a, miR-10b, miR-24, miR-145, miR-183, miR-197, miR-21, miR-26a-1, miR-27b, miR-411, miR-423, and miR-451a [19,24,27,34,35,36,39,40].

Platelet miRNAs are known to regulate platelet protein expressions [25,40,59]. The study by Kannan et al. (2009) found that miR-145 expression levels gradually declined from day two in response to storage at room temperature. It was revealed that as levels of this miRNA declined, its apoptosis-relevant targets (*F11R*, *CORT*, and *TNFRSF10B*) were identified upregulated until the end of the platelet life cycle under ex vivo conditions [24]. During the platelet storage process in blood banks, decreased miRNA abundance may occur due to shear stress and platelet activation, which are responsible for the release of miRNA-rich microparticles [8,42,44].

This stress is caused by aging due to the period of platelet storage for more than five days [19]. For example, miR-150-5p, miR-501-3p, miR-338-5p and miRNAs were quantified below the mean of relative expression in blood banked PC bags, indicating that the more pronounced the decrease in expression of these miRNAs, the greater is the deterioration of stored platelets [19]. Possibly, post-transcriptional modifications may influence miRNA biogenesis and stability during ex vivo storage, such as molecular changes in DICER1 reducing the number of miRNAs that strongly regulate platelet reactivity [24,32].

When miRNAs are transported in circulating MPs, they stabilize, especially after activation. MPs miRNA profiles have been found to differ from their source sites, indicating selective packaging of cell miRNAs for MPs [44]. In this review, we find it important not only to emphasize those studies that show platelet storage at different time intervals, as the main circumstance capable of altering miRNA expression profiles but also other circumstances that revealed different particularities responsible for these expression changes.

For example, a study using pathogen reduction (PR) systems for platelets stored in blood banks, which consists of using a psoralen-based Helinx technology designed to inactivate pathogenic nucleic acids through chemical mechanisms. This system affected the regulation of miRNAs and mRNAs (anti-apoptotic), activation, and platelet function. However, PR did not affect the synthesis or function of platelet miRNAs. On the other hand, of the miRNAs observed in this study, there was a reduction in miR-223 and let-7e levels, inducing activation, compromising platelet response to physiological agonists, resulting in the release of miRNAs through MPs [37,38]. One study reported the effects of cooling or freezing of residual platelets in plasma, leading to an increase in the number of activated platelet MPs that caused increased expression of miR-21 and miR-27b [39]. Thus, platelet miRNA profiles can be modified as a consequence of MPs activation-induced release. These miRNAs are listed in Table 1.

Studies on platelet storage show that apoptosis has been extensively studied to understand the causes of platelet cell damage under common storage conditions in standard blood banks, rendering them ineffective for transfusion [29,30,31]. Understanding the causes of these events requires a better understanding of the post-transcriptional regulation of apoptotic mRNA translation control that is a cause of PSL [24,30]. MiRNAs can silence important mRNAs involved in regulating platelet activation functions leading to structural modifications, phospholipid expression with increased procoagulant activity, microvesicle formation, reduction in mean platelet volume, ATP depletion with lactate synthesis, cytoskeleton proteins, and decreased mitochondrial membrane potential leading to apoptosis [3,12,17].

### 4.2. Increased miRNA Expression during Storage

Pontes et al. (2015) reported that approximately 22% of miRNAs experienced a reduction in their expression levels from day 1 to day 5, but increased after seven days of blood collection, most likely as a response to inhibiting stress-induced protein translation caused by aging in platelet storage of more than five days [7].

Analysis of the miRNA differential expression showed how the variation in the mean relative expression reflected the miRNA half-life instability on the fourth day of PC storage, coinciding with the time of onset of platelet storage lesions [19]. Measuring miRNA differential expression made it possible to identify PC bags of upregulated miRNAs that were expressed on all storage days and probably regulate genes that keep the platelet alive, activated, or aged [19]. These upregulated miRNAs can be very useful for selecting PC bags that still contain physiologically normal platelets suitable for transfusion in blood bank governance.

Differential expression analysis to identify upregulated miRNAs in PC bags stored for more than four days enables selection of PC bags that still contain physiologically normal platelets suitable for transfusion in blood banks. The durability of platelet physiology depends on the number of suboptimal health status (SHS) blood donors of which the PC bag is composed [60]. SHS is a public health problem in a population characterized by ambiguous health complaints, loss of vitality, chronic fatigue and low energy levels within a 3-month period. Individuals with SHS have a relative shortening of telomeres that may be associated with early platelet aging [61].

Some potential mechanisms are suggested to explain how miRNA expression may be increased in stored human platelets. Aging-associated miRNA precursors are converted to mature miRNAs by RNA editing enzymes [62]. RNA precursors can be cleaved into small RNA fragments with a regulatory ability to suppress protein translation in response to stress [63]. Another possibility is that cold-inducible RNA-binding protein, such as RBM3, regulating miRNA biogenesis at the DICER step, is stimulated during the process of being cooled to 4 °C and contributions to increased miRNA expressions [64]. These mechanisms seem reasonable because platelets have a rapid turnover of miRNAs [65] and expression levels of platelet miRNAs increase in a short time under marked stress [66].

This stress can be caused by aging caused by the platelet storage period for more than five days. Blood component platelets are more susceptible to oxidative stress as pointed out in mitochondrial stress tests, where it has been shown that the overall bioenergetic health of stored platelets is significantly lower than in fresh platelets [67]. Indeed, platelet mitochondrial DNA plays a key role in the regulation of apoptosis [30], and analysis of mitochondrial dysfunction of these cells reveals the loss of platelet quality during storage [29].

Studies have shown that specific miRNA expression levels such as let-7b/miR-16 [24], miR-326 [31], miR-191/miR-127/miR-320a [7], miR-548a [68], miR-570 [29], and miR-1304/miR-411/miR-432/miR-668/miR-939 [19] may increase at room temperature. This increase in miRNA expression in PC, possibly under severe stress for more than four days of storage, reinforces our argument that they are potential candidates for regulating platelet physiology [24,27] under these conditions and can still be used to select those PC bags that are viable for transfusion.

Let-7 family miRNAs were identified with an upregulated expression profile and acting on the regulation of platelet apoptosis stored in blood banks (Table 1). These miRNAs represent 48% of the content of these cell transcripts [56] and have been identified by regulating apoptosis-specific genes such as caspase-3 (CASP3) and a-isoform of apoptosis regulator (BAX) [24].

Similarly, miR-326/miR-145/miR-155/miR-150/miR-96/miR-25/miR-24/miR-16/miR-15/miR-7 plays a crucial role in platelet apoptosis during storage [30,31]. Upregulated miRNAs, primarily miR-1304-3p and miR-432-5p (Table 1), was also identified by regulating target genes from mitochondrial ATPase families, caspases, and proteins that act strongly on platelet aggregation and apoptosis [19]. In addition, increasing PC exposure time during storage favored decreased miR-21 and miR-155 expression levels on day 5, followed by increased miR-223, miR-3162, and let-7b (Table 1), which directly impacted the platelet aggregation pathway and, consequently, the PSL [38]. Figure 2 is a hypothetical diagram summarizing the role of miRNAs in severe stress apoptosis, which was based on previous studies adaptations [9,49,69].

During platelet storage, apoptosis is mainly triggered by shear stress and activation [49]. Platelet life span is regulated by the intrinsic apoptosis pathway (**1**) [69]. *Bcl-xl* (**a**) is the essential mediator of platelet survival that operates by restricting Bak and Bax (**b**) the critical facilitators of the death cell. BH3 primer proteins can inhibit and displace pro-survival proteins, thereby activating Bak and Bax to trigger mitochondrial outer membrane permeabilization (MOMP), leading to the release of cytochrome-c (**c**) and downstream assembly of Apaf. -1 and Caspase-9 in apoptosome, which leads to activation of effector caspases (Caspase-3/7) (**d**) and subsequent exposure to phosphatidylserine (PtdSer) (**e**). The death receptor-mediated extrinsic apoptosis pathway (**2**) [69] is activated by ligand binding to cell surface receptors, including Fas receptor, in addition to other apoptosis receptors. Caspase-8 is a critical member of the extrinsic pathway and its activation is followed by direct activation of effector Caspase. In some cell types, activation of the BH3 initiator protein Bid links the extrinsic and intrinsic pathways downstream of Caspase-8 activation. In (**3**), there is also increased expression of miRNAs that are capable of regulating members of the *Bcl-xl* family, families of mitochondrial ATPases, caspases, and proteins that act strongly on platelet aggregation [19,29]. MiRNAs can be processed by the Dicer associated with the TRBP protein by cleaving the two filaments near the hairpin loop to generate a duplex that is Argonaut-loaded (AGO), where selective incorporation of a mature miRNA chain that will be loaded into the RNA-induced silencing complex (RISC) to bind to target mRNAs [9]. (**4**) The microvesicular pathway: miRNA uptake by microvesicles after activation or during apoptosis [46]. *BH3-only initiator proteins (BIM, PUMA, BAD, NOXA, BIK, HRK, and BMF), Mitochondrial Outer Membrane Permeabilization (MOMP), Phosphatidylserine (PtdSer), Xk-related protein 8 (Xkr8): Has been identified as the enzyme catalyzing caspase-dependent lipid scrambling, Transactivation responsive RNA binding protein (TRBP), and RNA-induced silencing complex (RISC).

It is not yet clear whether the observed miRNA differential expression is universally due to the degradation of miRNAs as a result of storage in all types of stored PC bags. One study shows that cold storage conditions modify miRNA expressions. Increased levels of miR-20a, miR-10a, miR-16-2 and miR-223 (Table 1) correlated with platelet quality under specific storage conditions [20]. This feature is important because anucleated cells, such as platelets, do not actively synthesize miRNAs, indicating that PC degradation may be a major factor in the increase in miRNAs. Therefore, we believe that differential expression is capable of measuring miRNA level during platelet storage and correlating the variation of miRNA level with platelet quality under specific storage conditions. This is a way to clarify what leads to miRNA degradation in PC bags.

### 4.3. Analytical Methods for Quality Assessment of Platelet Concentrates Stored in Blood Banks

Some analytical laboratory methods can be applied to assess process that can verify which PC bags are in good condition for transfusion, increasing the reliability of stored platelet quality and reducing waste. We know that blood banks in most countries discard all unused PC bags after five days of storage, making PC bags scarce on the sixth day of the blood center. However, many of these PC bags contain non-activated or functional platelets that can be stored for another day in the absence of new PC bags.

A simple methodology that can be applied uses microRNA specific primers for detection of reverse-transcribed microRNAs [70], making them highly reliable and reproducible. All of these microRNAs should be extracted, purified from qPCR inhibitors to avoid sources of pre-analytical variations, and then quantified from a small sample of platelet concentrate by standard methodologies.

A first analytical process compares the relative expression of miR-127 and miR-320a microRNAs identified by Pontes et al. (2015) on platelets stored in a blood bank. In this type of analysis, when miR-127 has an expression of less than 80% of miR-320a expression in a platelet concentrate bag, it means that there are storage lesions in this blood component and suggests blockage of the platelet concentrate bag. We propose that this relationship between miR-127 and miR-320a can be used in blood banks as part of stored platelet quality control and PC bags can be tested at any time. On the other hand, the platelet concentrate bag will be considered suitable for transfusion when it has one of the following possibilities: (i) miR-127 expression > miR-320a, (ii) equal expression between miR-127 and miR-320, and (iii) the miR-127 expression difference from miR-320a is less than 20%.

The second analytical process can use the upregulated miRNAs (miR-1304-3p, miR-432-5p, miR-411-5p, miR-668-3p, and miR-939-5p) identified with differential expression analysis by Maués et al. (2018) who may compose a specific strategic panel to conduct routine clinical trials of healthy blood bank donors. This type of test uses analysis of the differential expression of these miRNAs comparing the fourth-day storage bags with the first day bags (used as platelet quality control).

The PC bags will be considered suitable for transfusion when in these bags the expression of upregulated miRNAs is greater than 80% of the expression of these miRNAs. On the other hand, when the expression value of these miRNAs is less than 80%, it indicates that the bags contain platelet with storage lesion. This test can also help differentiate a minority of non-injured PC bags that may be suitable for transfusion.

Analytically, miRNAs were found to be stable for up to 48 h in human biofluids, even when stored at room temperature [70]. However, more accurate measurement of miRNAs on PC bags has some challenges that need to be considered. The first methodology employs the use of only two microRNAs, to avoid limitations that may make this methodology challenging, miRNAs are very short and have highly divergent sequences, with a wide variation of GC content that may favor the different hybridization properties between different miRNA sequences [70] which can be challenging to measure only two miRNAs simultaneously on many PC bags.

To reduce the limitations that may occur in the first methodology, the second employs a specific group of upregulated miRNAs that were identified in all seven days of PC storage. Also, these miRNAs can be tested on the fourth day, that is, the day before those miRNAs that are tested only on the fifth day. Therefore, the “method for determining the quality of platelet concentrates stored in a blood bank” is the first proposed process for testing the quality of PC bags for molecular biology and can be done quickly because it uses reliable microRNAs and results.

## 5. Conclusions

In conclusion, miRNAs are an effective tool for identifying the molecular mechanism by which storage conditions result in functional differences in human platelets. As thousands of blood products are transfused each year and many lives are directly affected by transfusion, tools and strategies that evaluate the quality of PC bags need to be increasingly improved to ensure the safety and effectiveness of patients’ health. PC transfusion practices can be suboptimal in many ways and vary widely between different hospitals and clinics. Considering the number of suboptimal healthy blood donors from which the PC bag is composed, the identification of miRNA profiles to measure PC quality is possible. Therefore, miRNAs can be used to measure the quality of storage PC for more than 5 days, to identify PC bags with platelet injury, and distinguish those with functional platelets.

## Figures and Tables

**Figure 1 cells-08-01256-f001:**
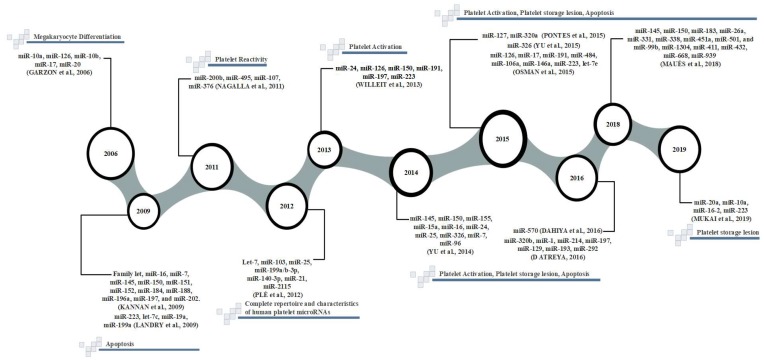
Timeline of major platelet miRNA publications and their relationship to platelet storage lesions.

**Figure 2 cells-08-01256-f002:**
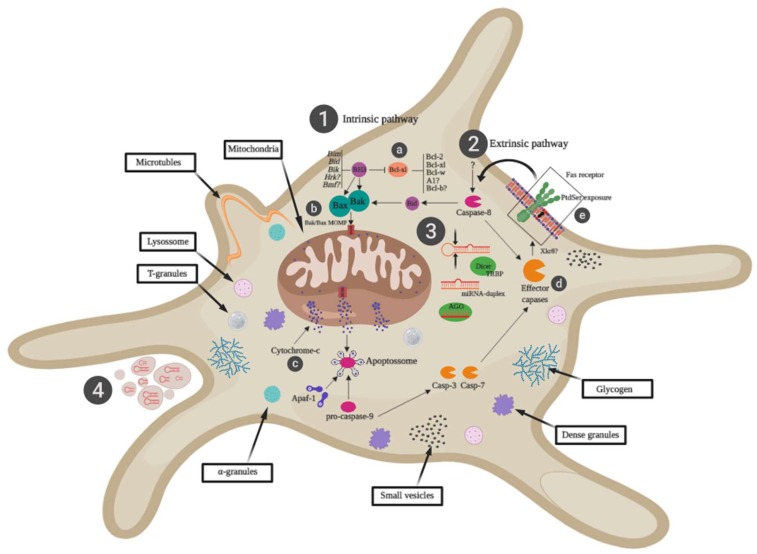
The role of miRNAs in platelet apoptosis.

**Table 1 cells-08-01256-t001:** A list of miRNAs and their main functions in platelets.

MicroRNAs *	Study Approach	Functions	References
miR-127, miR-320a.	Differential Expression Analysis. Platelet Storage.	Platelet activation.	[7] ^a.^
miR-145, miR-150, miR-183, miR-26a, miR-331, miR-338, miR-451a, miR-501, and miR-99b, miR-1304, miR-411, miR-432, miR-668, miR-939.	Differential Expression Analysis. Platelet Storage.	Platelet activation. Storage Lesion. Apoptosis.	[19] ^a.^
miR-20a, miR-10a, miR-16-2, miR-223.	Differential Expression Analysis. Platelet Storage.	Storage Lesion.	[20] ^a.^
miR-10a, miR-126, miR-10b, miR-17, miR-20	Differential Expression Analysis.	Megakaryocyte Differentiation	[23] ^b^
Family let (-7a, -7b, -7c, 7e, -7f, -7g, -7i). mir-223, let-7c, miR-19a, miR-140, miR-15 miR-16, miR-339, miR-365, miR-495, miR-98, miR-361, miR-200b, miR-495, miR-107, miR-376.	Differential Expression Analysis. Platelet Storage. Platelet microparticles.	Platelet Reactivity. Platelet Regulatory Pathways Apoptosis.	[24] ^a.^ [25] ^b.^ [26] ^a.^ [27] ^b.^ [28] ^a.^
miR-570.	Differential Expression Analysis. Platelet Storage.	Storage Lesion. Apoptosis.	[29] ^b.^
miR-326, miR-96, miR-16, miR-155, miR-150, miR-7, miR-145, miR-24, miR-25, miR-15a.	Differential Expression Analysis. Platelet Storage.	Platelet activation. Storage Lesion. Apoptosis.	[30] ^c.^
miR-326.	Differential Expression Analysis. Platelet Storage.	Storage Lesion. Apoptosis.	[31] ^c.^
miR-326, miR-128, miR-331, miR-500.	Differential Expression Analysis. Platelet Storage.	Platelet Reactivity.	[32] ^a,b.^
miR-320b, miR-1, miR-214, miR-197, miR-129, miR-193, miR-292.	Differential Expression Analysis. Platelet Storage.	Storage Lesion.	[33] ^b.^
miR-200b, miR-495, miR-107, miR-376.	Differential Expression Analysis.	Platelet Reactivity.	[34] ^a.^
miR-16, miR-22, miR-185, miR-320b, miR-423.	Differential Expression Analysis. Platelet microparticles.	Platelet activation.	[35] ^a.^
miR-223, miR-191, miR-197, miR-24, miR-21, miR-126, miR-150.	Support Vector Machines (SVM). Platelets, microparticles, platelet-rich. plasma, platelet-poor plasma and serum.	Platelet activation.	[36] ^c^
miR-126, miR-17, miR-191, miR-484, miR-106a, miR-146a, miR-223, let-7e.	Differential Expression Analysis. Platelet Storage.	Platelet activation. Apoptosis.	[37] ^b.^
miR-27b, miR-126, miR-21, miR-451.	Differential Expression Analysis. Platelet Storage. Platelet microparticles.	Platelet activation.	[38] ^c.^
miR-21, miR-155, miR-223, miR-3126, let-7b.	Differential Expression Analysis. Platelet Storage.	Platelet Aggregation. Storage Lesion. Apoptosis.	[39] ^b.^
miR-10a, miR-10b.	Differential Expression Analysis.	Megakaryocyte Differentiation	[40] ^c^

^a^ RNA-seq, ^b^ Microarrays, and ^c^ Quantitative PCR. ***** Most cited MicroRNAs in the articles: miR-223 (five studies), miR-126 (four studies), miR-10a/miR-150/miR-16/miR-21/miR-326/miR-495 (three studies), let-7b/let-7c/let-7e/miR-107/miR-10b/miR-145/miR-155/miR-17/miR-191/miR-197/miR-200b/miR-24/miR-331 and miR-376 (two studies).

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
