# Peer review of "MicroRNAs as a Potential Quality Measurement Tool of Platelet Concentrate Stored in Blood Banks—A Review"

_cells, 2019, doi:10.3390/cells8101256_

Round 1
Reviewer 1 Report
This short review article by da Silva Maués summarizes current research on platelet microRNAs and their significance as quality markers for storage of platelet concentrates in blood banks. The topic is important and timely and the authors have highlighted the importance of platelet quality and safety in transfusion medicine, where microRNAs could be potentially useful as quality biomarkers. There are however several concerns, which unfortunately raise doubts on whether this review paper can give us any clarity on the value of microRNAs as useful biomarkers in platelet transfusion.
Major concerns:
Authors have sadly failed to critically review previous research and give a balanced, state-of-the-art discussion that clearly gives the reader information on where we are today and what remains to be done in the future. In many sections of the text, phrases starting with “Studies have shown that….” are not followed by any critical discussion. Authors basically repeat what previous studies already have said, which is already known and can be found in the literature. Authors should instead discuss the strengths and weaknesses of those studies and what the future perspectives should be. Authors uncritically and repeatedly mention that the Let-7 family microRNAs represent 48% of the total platelet microRNome, citing the study by Plé et al. (Plos One, 2012). However, there are several other studies that did not find similar results, and authors should not focus too much on that single study and ignore other studies with different outcome. Authors should be aware that different studies using different methods can come to different conclusions and that not one single study represents the whole “truth”. Page 8, first paragraph, line 207. Two studies are cited (48 and 55) but only one is relevant to the claim made by the authors. The study in reference 55 did not investigate miR-223 or Let-7e, or even any other microRNA at all. Authors should be careful with their references and only cite those studies that support their claim. Figure 2. It is unclear whether the diagram presented in fig. 2 is a hypothesis made by the authors, or whether it is based on previous research. Authors should be clear on this either by stating that this is their hypothesis or by citing other previous work in the figure legend. In several sections of the text, authors write “conditional exclusion system” and cite other studies but fall short to explain for the reader what this means. Similarly, “action transport” is mentioned in several places without any further explanation. Authors should avoid “flashy” and “trendy” words that are difficult to comprehend and that do not add any value to the text. Page 5, last paragraph, line 170-171. Authors write the following text but do not cite any study: “Studies have described that there is a strong correlation between the platelet transcriptome and its proteomic profile, which maintains that the de novo platelet translation capabilities do exist.” Authors should also know that this is a highly controversial claim. For instance, while McRedmond et al. (Mol Cell Proteomics. 2004) found that 69% of secreted proteins were detectable at the mRNA level in platelets, it doesn’t mean that all these proteins were de novo synthetized in platelets. As authors mention in the text, platelets have no nuclei and all of their transcriptome and overwhelmingly most of their protein contents are derived from their parent megakaryocyte cell. Only a very limited number of proteins have been so far reported to be newly translated in platelets, mostly by the labs of Weyrich and Zimmerman, and mostly not confirmed by other researchers.Minor issues
The text contains several typos, including but not limited to the following:
Page 3, paragraph 3, line 93-94: “miRNAs” are repeated 4 times. Page 8, paragraph 3: “Upregulated miRNAs probably regulate genes that keep the platelet alive but activated or aged”. This sentence should re-written and clarified. Page 9, paragraph 4, line 252: “Studies have show” should be changed to “Studies have shown”. Page 10, paragraph 1, line 265: “iddentified” should be changed to “identified”. Page 4, paragraph 3, line 123: “miRnoma” should be change to “miRNome”Author Response
Reviewer 1
This short review article by da Silva Maués summarizes current research on platelet microRNAs and their significance as quality markers for storage of platelet concentrates in blood banks. The topic is important and timely and the authors have highlighted the importance of platelet quality and safety in transfusion medicine, where microRNAs could be potentially useful as quality biomarkers. There are however several concerns, which unfortunately raise doubts on whether this review paper can give us any clarity on the value of microRNAs as useful biomarkers in platelet transfusion.
Major concerns:
Authors have sadly failed to critically review previous research and give a balanced, state-of-the-art discussion that clearly gives the reader information on where we are today and what remains to be done in the future. In many sections of the text, phrases starting with “Studies have shown that….” are not followed by any critical discussion. Authors basically repeat what previous studies already have said, which is already known and can be found in the literature. Authors should instead discuss the strengths and weaknesses of those studies and what the future perspectives should be.
Authors uncritically and repeatedly mention that the Let-7 family microRNAs represent 48% of the total platelet microRNome, citing the study by Plé et al. (Plos One, 2012). However, there are several other studies that did not find similar results, and authors should not focus too much on that single study and ignore other studies with different outcome. Authors should be aware that different studies using different methods can come to different conclusions and that not one single study represents the whole “truth”.
Page 2
L53-L55: The paragraph has been rewritten and the appropriate reference inserted.
In human platelets, one of the most abundant miRNA families has been described as members of the let-7 family, which account for 48% of platelet miRNA content [29]. The existence of let-7 family members (let-7a, -7c, -7e, -7f, -7g, and -7i) (Table 1) has been previously confirmed on platelets during storage [25]. Subsequent studies also showed different results identifying the miR-127 / miR-320a [7], miR-548 [30] family that were shown to be abundant during platelet storage. These findings show that different sequencing technologies, combined with bioinformatics tools and pipelines, yield different results and show how the platelet miRNA repertoire is relatively diverse and complex.
Page 8, first paragraph, line 207. Two studies are cited (48 and 55) but only one is relevant to the claim made by the authors. The study in reference 55 did not investigate miR-223 or Let-7e, or even any other microRNA at all. Authors should be careful with their references and only cite those studies that support their claim.
L257-L264: This paragraph has been better clarified. To this it was added in place of the paragraph: In vitro studies show that platelets treated for pathogen reduction exhibited a decrease in miR-223 and let-7e of MPs-activated platelets [49,56].
For example, a study using pathogen reduction (PR) systems for platelets stored in blood banks, which consists of using a psoralen-based Helinx technology designed to inactivate pathogenic nucleic acids through chemical mechanisms. This system affected the regulation of miRNAs and mRNAs (anti-apoptotic), activation and platelet function. However, PR did not affect the synthesis or function of platelet miRNAs. On the other hand, of the miRNAs observed in this study, there was a reduction in miR-223 and let-7e levels, inducing activation, compromising platelet response to physiological agonists, resulting in the release of miRNAs through MPs [56].
L264: Reference [57] has been excluded from the manuscript.
Figure 2. It is unclear whether the diagram presented in fig. 2 is a hypothesis made by the authors, or whether it is based on previous research. Authors should be clear on this either by stating that this is their hypothesis or by citing other previous work in the figure legend.
We inserted the references in the caption of Figure 2. L340: [44], L341: [72], L347: [72], L353: [19,30], L356: [9], L357: [38].
In several sections of the text, authors write “conditional exclusion system” and cite other studies but fall short to explain for the reader what this means. Similarly, “action transport” is mentioned in several places without any further explanation. Authors should avoid “flashy” and “trendy” words that are difficult to comprehend and that do not add any value to the text.
Page 5:
L138-L145: The paragraph has been rewritten. We remove the term conditional exclusion since Rowley et al. (2016) used a molecular modification system to delete Dicer1 in murine platelets.
In order to shape mRNA profile and murine platelet function, Rowley et al. (2016) used a molecular alteration system to delete Dicer1, as described in their study [33], which resulted in increased expression of αIIbβ3 integrins on the surface of these cells, leading to increased platelet reactivity [33]. The authors suspected that miRNAs would be regulating these integrins and tested their hypotheses using 3 'untranslated region luciferase reporter assays confirming modulation of these mRNAs by miRNAs (miR-326, miR-128, miR-331, and miR-500), table 1, in megakaryocyte and potentially in platelets in the bloodstream or during storage [34].
L251-L256: Paragraph has been rewritten: When miRNAs are transported in circulating MPs, they stabilize for their action transport, especially after activation [36,37]. It has been found that MPs miRNA profiles may differ from their source sites, indicating selective cell miRNA packaging for MPs [36].
When miRNAs are transported in circulating MPs, they stabilize, especially after activation. MPs miRNA profiles have been found to differ from their source sites, indicating selective packaging of cell miRNAs for MPs [36].
Page 5-6:
L153-L158: The paragraph has been rewritten. We explain MPs better and insert a new reference from a study by GIBBINGS et al. (2009) describing a miRNA uptake pathway by MPs, which reinforces our hypothesis about the release of miRNA-rich microparticles during activation.
Release of miRNA-rich microparticles (MPs) is mainly triggered by factors such as shear stress and activation, which may also modify miRNA expression profiles during storage in blood banks [36,37]. A study by GIBBINGS et al. (2009) describes a miRNA uptake pathway by MPs, which reinforces our hypothesis about the release of miRNA-rich microparticles during activation. The main pathway called exosomal occurs through the lumen sprouting of multivesicular bodies that are released by exocytosis [38].
Page 5, last paragraph, line 170-171. Authors write the following text but do not cite any study: “Studies have described that there is a strong correlation between the platelet transcriptome and its proteomic profile, which maintains that the de novo platelet translation capabilities do exist.” Authors should also know that this is a highly controversial claim. For instance, while McRedmond et al. (Mol Cell Proteomics. 2004) found that 69% of secreted proteins were detectable at the mRNA level in platelets, it doesn’t mean that all these proteins were de novo synthetized in platelets. As authors mention in the text, platelets have no nuclei and all of their transcriptome and overwhelmingly most of their protein contents are derived from their parent megakaryocyte cell. Only a very limited number of proteins have been so far reported to be newly translated in platelets, mostly by the labs of Weyrich and Zimmerman, and mostly not confirmed by other researchers.
L215-L221: This paragraph has been completely rewritten in accordance with the reviewer's recommendation. Studies have described that there is a strong correlation between the platelet transcriptome and its proteomic profile, which maintains that the de novo platelet translation capabilities do exist. In addition to maintaining processing of precursor and mature miRNAs [22,49], suggesting the possibility of post-transcriptional regulation of gene expression within platelets during storage conditions [25].
Platelets are able to synthesize a limited amount of proteins from resident mRNAs [48], along with other processes such as modulating miRNA expression [49], in platelets extensive pre-mRNA maturation occurs as responses to stimuli activators, representing a mechanism for post-transcriptional control of the proteome arrangement in these anucleated cell fragments. Still, studies have shown that platelet proteomic data correlated well with transcriptome, as about 69% of secreted proteins were detected at the mRNA level, meaning that not all proteins were synthesized again [50,51].
Minor issues
The text contains several typos, including but not limited to the following:
Page 3, paragraph 3, line 93-94: “miRNAs” are repeated 4 times.
Page 3
L95-L97: The paragraph has been corrected as suggested by the reviewers.
In this study, miRNAs with a downregulated profile such as (miR-10a, miR-126, miR-106, miR-10b, miR-17 and miR-20) involved in megakaryocytopoiesis were identified [23] (Table 1).
L279: Title has been rewritten: Platelets Concentrate increased expression profiles of miRNA under severe stress
4.2 Increased miRNA expression during storage
L280-L283: Paragraph has been modified: Pontes et al. reported that approximately 22% of miRNAs experience a reduction in their expression levels from day 1 to day 5, while other miRNAs may increase even after seven days of blood collection. Upregulated miRNAs probably regulate genes that keep the platelet alive but activated or aged [19].
Pontes et al. reported that approximately 22% of miRNAs experienced a reduction in their expression levels from day 1 to day 5, but increased after seven days of blood collection, most likely as a response to inhibiting stress-induced protein translation caused by aging. platelet storage for more than five days [7].
L284-L291: Paragraph has been rewritten.
Analysis of the miRNA differential expression showed how the variation in the mean relative expression reflected the miRNA half-life instability on the fourth day of PC storage, coinciding with the time of onset of platelet storage lesions [19]. Measuring miRNA differential expression made it possible to identify pockets of upregulated miRNAs that were expressed on all storage days and probably regulate genes that keep the platelet alive, activated or aged [19]. These upregulated miRNAs can be very useful for selecting PC pouches that still contain physiologically normal platelets suitable for transfusion in blood bank governance.
Page 9, paragraph 4, line 252: “Studies have show” should be changed to “Studies have shown”.
L287: Studies have show (Studies have shown)
Page 10, paragraph 1, line 265: “iddentified” should be changed to “identified”.
L264-L265
was also identified by regulating target genes from mitochondrial ATPase families, caspases, and proteins that act strongly on platelet aggregation and apoptosis [19].
Page 4, paragraph 3, line 123: “miRnoma” should be change to “miRNome”
L132-L133
The first complete miRNA sequencing (miRNome) of blood bank stored platelet concentrate was obtained with NGS and revealed a gradual decrease of the most abundant miRNAs in PC.
L368-L371
Therefore, we believe that differential expression is capable of measuring miRNA level during platelet storage and correlating the variation of miRNA level with platelet quality under specific storage conditions. This is a way to clarify what leads to miRNA degradation in PC bags.
Reviewer 2 Report
MicroRNAs as a Tool for Measuring the Quality of Platelets Stored in Blood Banks: A Review.
cells-583667
This review deals with microRNAs in stored platelets, and describes what is known about platelet miRNA expression in stored platelets and the factors impacting the expression of platelet miR's.
The review is in general well described with an adequate amount of references, covering relevant aspects of microRNA biology in stored platelets.
I have a few comments
The title declares that we are about to learn to use microRNAs as a quality tool. However, the review offers no suggestions on how to use microRNA's as a quality tool. The paper should include this, and also describe some perspectives on how such a quality step could be implemented in the laboratory.
Some sentences did not give much meaning, please rephrase:
L25: Studies also show that PC quality biomarkers such as miR-126, miR-223, miR-150,26 miR-21, miR-326, let-7b, let-7e, miR-10a, miR-145, miR- 155, miR-15a, miR-16, miR-17, miR-191, miR-27 197, miR-24, miR-320b, miR-331. They are very important in blood bank governance to identify PC 28 bags with platelet injury.
L52: Studies have shown that a proposed mechanism based on the molecular regulation of platelet microRNAs (miRNAs) has been shown to be relevant for storage lesions, since platelets, when stored, continue to translate mRNA proteins
L93: In addition, miRNAs miRNAs miRNA regulating miRNAs miR-10a, miR-126, miR-106, miR-10b, miR-17 and miR-20 were first identified
L137: interacted with the mitochondrial ATPase subunit (ATP5L) that encodes mRNA in stored platelets
L 205: In vitro studies show that platelets treated for pathogen reduction exhibited a decrease in miR-223 and let-7e of MPs-activated platelets
L241: is markedly stimulated during the process of being cooled to 4°C and contributions to increased miRNA expressions
L256: suggests downregulated regulation of its targets
Other comments
Ll140: The abbreviation MP is used for microparticles throughout the paper, while it is first defined as "activated platelet microparticle-derived miRNAs (MPs)"
L155: The heading is misleading. Decreasing can not be used in the context of expression profiles. A profile will typically contain increased, unchanged and decreased expression of miRNA's.
L223: Misleading heading, as above
Table 1, L180: Several places with non-english text
L234: "Individuals with SHS have a relative shortening of telomeres that can lead to early platelet aging" Is the causal relationship between telomere length and platelet aging evident? Otherwise it should be stated as an association.
Author Response
Reviewer 2
MicroRNAs as a Tool for Measuring the Quality of Platelets Stored in Blood Banks: A Review.
cells-583667
This review deals with microRNAs in stored platelets, and describes what is known about platelet miRNA expression in stored platelets and the factors impacting the expression of platelet miR's.
The review is in general well described with an adequate amount of references, covering relevant aspects of microRNA biology in stored platelets.
I have a few comments
The title declares that we are about to learn to use microRNAs as a quality tool. However, the review offers no suggestions on how to use microRNA's as a quality tool. The paper should include this, and also describe some perspectives on how such a quality step could be implemented in the laboratory.
Some sentences did not give much meaning, please rephrase:
L25: Studies also show that PC quality biomarkers such as miR-126, miR-223, miR-150,26 miR-21, miR-326, let-7b, let-7e, miR-10a, miR-145, miR- 155, miR-15a, miR-16, miR-17, miR-191, miR-27 197, miR-24, miR-320b, miR-331. They are very important in blood bank governance to identify PC 28 bags with platelet injury.
Page 1
L24-L28: We have modified this paragraph suggested by the reviewers. And consider the following:
The major studies listed in this review identified miRNAs associated with important platelet functions that are relevant in clinical practice as quality biomarkers of PC such as miR-126, miR-223, miR-150, miR-21, miR-326, let. -7b, let-7e, miR-10a, miR-145, miR-155, miR-15a, miR-16, miR-17, miR-191, miR-197, miR-24, miR-320b, miR-331. These miRNAs can be used in blood banks to identify platelet injury PC bags.
L52: Studies have shown that a proposed mechanism based on the molecular regulation of platelet microRNAs (miRNAs) has been shown to be relevant for storage lesions, since platelets, when stored, continue to translate mRNA proteins
Page 2
L53-L55: The paragraph has been rewritten and the appropriate reference inserted.
A proposed mechanism based on the molecular regulation of microRNAs (miRNAs) has been shown to be relevant for understanding storage lesions, as platelets when stored, continue to translate mRNA proteins [7]. The biological role of miRNA is mainly linked to its ability to act together and mediate sequence-specific regulation by repressing or degrading mRNAs at specific non-translated binding sites [8,9]. This review addresses the major advances in the identification of platelet miRNAs that are important in the field of blood components transfusion and associating the role of miRNAs as a tool to measure the quality of platelets stored in blood banks.
L93: In addition, miRNAs miRNAs miRNA regulating miRNAs miR-10a, miR-126, miR-106, miR-10b, miR-17 and miR-20 were first identified
Page 3
L95-L97: The paragraph has been corrected as suggested by the reviewers.
In this study, miRNAs with a downregulated profile such as (miR-10a, miR-126, miR-106, miR-10b, miR-17 and miR-20) involved in megakaryocytopoiesis were identified [23] (Table 1).
L137: interacted with the mitochondrial ATPase subunit (ATP5L) that encodes mRNA in stored platelets
Page 5:
L146-L152: The paragraph has been rewritten.
In many countries with few blood donors, it is common during PC storage, even under optimal storage conditions, platelets begin to lose their quality, which is due to mitochondrial dysfunction [35]. Using bioinformatics analyzes one study showed ATP5L encoding mRNA as a potential target for miR-570-3p, which was identified with high levels during platelet storage, being able to clarify the mechanisms of mitochondrial dysfunction of this cell by miRNA regulation by mRNAs. The miR-570 (Table 1) have been proposed as a reliable biomarker for the quality and viability of stored platelets [30].
L 205: In vitro studies show that platelets treated for pathogen reduction exhibited a decrease in miR-223 and let-7e of MPs-activated platelets
L257-L264: This paragraph has been better clarified. To this it was added in place of the paragraph: In vitro studies show that platelets treated for pathogen reduction exhibited a decrease in miR-223 and let-7e of MPs-activated platelets [49,56].
For example, a study using pathogen reduction (PR) systems for platelets stored in blood banks, which consists of using a psoralen-based Helinx technology designed to inactivate pathogenic nucleic acids through chemical mechanisms. This system affected the regulation of miRNAs and mRNAs (anti-apoptotic), activation and platelet function. However, PR did not affect the synthesis or function of platelet miRNAs. On the other hand, of the miRNAs observed in this study, there was a reduction in miR-223 and let-7e levels, inducing activation, compromising platelet response to physiological agonists, resulting in the release of miRNAs through MPs [56].
L264: Reference [57] has been excluded from the manuscript.
L241: is markedly stimulated during the process of being cooled to 4°C and contributions to increased miRNA expressions
L256: suggests downregulated regulation of its targets
This increase in miRNA expression in PC, possibly under severe stress for more than four days of storage, reinforces our argument that they are potential candidates for regulating platelet physiology [25,28] under these conditions and can still be used to select those PC bags that are viable for transfusion.
Other comments
Ll140: The abbreviation MP is used for microparticles throughout the paper, while it is first defined as "activated platelet microparticle-derived miRNAs (MPs)"
Page 5-6:
L153-L158: The paragraph has been rewritten. We explain MPs better and insert a new reference from a study by GIBBINGS et al. (2009) describing a miRNA uptake pathway by MPs, which reinforces our hypothesis about the release of miRNA-rich microparticles during activation.
Release of miRNA-rich microparticles (MPs) is mainly triggered by factors such as shear stress and activation, which may also modify miRNA expression profiles during storage in blood banks [36,37]. A study by GIBBINGS et al. (2009) describes a miRNA uptake pathway by MPs, which reinforces our hypothesis about the release of miRNA-rich microparticles during activation. The main pathway called exosomal occurs through the lumen sprouting of multivesicular bodies that are released by exocytosis [38].
L155: The heading is misleading. Decreasing can not be used in the context of expression profiles. A profile will typically contain increased, unchanged and decreased expression of miRNA's.
4.1 Downregulated miRNAs on stored PC bags
L223: Misleading heading, as above
L279: Title has been rewritten: Platelets Concentrate increased expression profiles of miRNA under severe stress
4.2 Increased miRNA expression during storage
Table 1, L180: Several places with non-english text
Page 8
Table 1: All modifications and corrections suggested by the reviewers were made. These new changes are marked and explained in the text.
L234: "Individuals with SHS have a relative shortening of telomeres that can lead to early platelet aging" Is the causal relationship between telomere length and platelet aging evident? Otherwise it should be stated as an association.
L298-L299: Paragraph has been rewritten.
Individuals with SHS have a relative shortening of telomeres that may be associated with early platelet aging [64].
The title declares that we are about to learn to use microRNAs as a quality tool. However, the review offers no suggestions on how to use microRNA's as a quality tool. The paper should include this, and also describe some perspectives on how such a quality step could be implemented in the laboratory.
Some sentences did not give much meaning, please rephrase:
L373: New inserted paragraph offers suggestions on how to use microRNA as a quality tool. The document included this paragraph describing the two "methods for analyzing the quality of platelet concentrates stored in a blood bank." And perspectives on how this methodology can be implemented in the laboratory.
4.3 Analytical methods for quality assessment of platelet concentrates stored in blood banks
Round 2
Reviewer 1 Report
Authors have made the necessary changes and significantly improved the paper. I am happy with the revised version of the manuscript.